# Personalised immunotherapy in sepsis: a scoping review protocol

Marleen A Slim ,[1,2] Niels van Mourik,[1,2] Joanna C Dionne ,[3,4,5,6]
Simon J W Oczkowski,[3,4,5] M G Netea,[7,8] Peter Pickkers,[9]
Evangelos J Giamarellos-Bourboulis ,[10] Marcella C A Müller,[1]
Tom van der Poll,[2,11] W Joost Wiersinga,[2,11] Alexander P J Vlaar,[1]
Lonneke A van Vught[1,2]

MAS and NvM contributed equally.

For numbered affiliations see end of article.

**Correspondence to**
Marleen A Slim;
m.a.slim@amsterdamumc.nl

## ABSTRACT

**Introduction** Sepsis, a life-threatening organ dysfunction syndrome occurring in the context of severe infections, remains a major burden on global health with high morbidity and high mortality rates. Despite recent advances in the understanding of its pathophysiology, the treatment of sepsis remains supportive of nature with few interventions specifically designed for treating this complex syndrome. The focus of sepsis trials has increasingly shifted towards targeting excessive inflammation and immunosuppression using immunomodulatory agents. However, it remains uncertain how to identify patients that could benefit from such treatment, whether treatments can be tailored to an individual's immune profile, or at which stage of the disease the intervention should be initiated. In this scoping review, we provide a comprehensive overview of current available literature on immunostimulatory and immunosuppressive therapies against sepsis.

**Methods and analysis** The aim of this scoping review is to describe and summarise current literature evaluating immunotherapy in adult patients with sepsis. The review will be performed using the framework formulated by Arksey and O'Malley. A comprehensive literature and study collection will be executed by searching PubMed, Embase, Cochrane CENTRAL and ClinicalTrials.gov to identify clinical trials and cohort studies concerning immunotherapy in adult patients with sepsis. Screening will be performed independently and in duplicate by two reviewers who will also independently extract data into prespecified spreadsheets. We will summarise evidence in tabular format with descriptive statistics. The reported evidence will convey knowledge on the types of immunotherapies studied, and currently being studied, in adult patients with sepsis.

**Ethics and dissemination** Approval from a medical ethics committee is not required. Once completed, the review will be submitted for publication in a peer-reviewed journal. These results will be of value to clinicians and researchers with an interest in advancing sepsis care.

## STRENGTHS AND LIMITATIONS OF THIS STUDY

⇒ This will be the first review using the scoping review methodology to systematically summarise which immunotherapies have been, and are currently being, studied in sepsis, this will highlight the gaps in in the field of immunotherapy in sepsis.
⇒ In this review, we will not only focus on immunotherapy alone, but also address studies that use a personalised approach to immunotherapy.
⇒ In this review, we will adhere to Preferred Reporting Items for Systematic Reviews and Meta-Analyses Scoping Review guidelines, conducting abstract screening, full-text screening and data extraction in independent duplication to facilitate greater inter-reviewer reliability.
⇒ A potential limitation of this review lies within the heterogeneity between trials; sepsis is a heterogeneous syndrome and trials use trial-specific inclusion and exclusion criteria, leading to possible heterogeneity within the population studied.

global disease burden of sepsis remains high, with 48.9 million cases annually and almost 20% of all global deaths are sepsis related.[3] Over the past few decades, significant progress in the understanding of the pathophysiology of sepsis has been made. Early diagnosis, timely antibiotic treatment and organ support are pivotal and are associated with a decrease in mortality rates.[2] Despite numerous efforts to date, there are few therapies which target the immunological basis of this life-threatening condition.[4] So far, treatment of sepsis mainly revolves around supportive care; administering antibiotics, fluids and vasopressors or initiating organ support.

During sepsis, the host response may be altered in multiple ways, explaining the highly heterogeneous clinical presentation, treatment responses and prognosis.[5] While the main pathophysiological cause of sepsis is often thought to be an excessive inflammatory

## INTRODUCTION
### Rationale

Sepsis is defined as a life-threatening organ dysfunction syndrome caused by a dysregulated host response to an infection.[1 2] The

BMJ

immune response to an infection, it is now understood that a secondary immunosuppressive response often occurs concurrently.[4–6] Modulating the immune response to infection represents a promising treatment target in sepsis. However, over 100 clinical trials investigating the use of immunomodulatory agents in sepsis have been unsuccessful in demonstrating a benefit of immuno-modulatory therapy.[4 7] One reason these trials may not have demonstrated a benefit is the use of a 'one-size-fits-all' approach to immunotherapy. Sepsis is a clinically heterogeneous disease syndrome, with individual patients demonstrating varied clinical responses to infections. This suggests that personalised immunomodulatory treatment, tailored to an individual patient's immune profile, may be a better approach to treating patients suffering from sepsis.[8] Trials using personalised immunotherapy could increase the number of patients in the study with the potential to benefit from the administered treatment, and minimise the number of patients exposed to treatments which are unlikely to help or even cause harm.[8] The first step towards implementation of personalised medicine using immunomodulatory treatment is to provide a structured and in-depth overview of currently available evidence on immunotherapy in the treatment of sepsis.

## Study aim

The purpose of this scoping review is to describe and summarise current literature evaluating immunotherapy in adult patients with sepsis and the extent to which these trials used a personalised medicine approach.

## METHODS AND ANALYSIS

This study will be a scoping review performed using previously defined methods.[9 10] The model by Arksey and O'Malley defines a five-stage methodological framework, which includes: (1) identifying the research question, (2) identifying relevant studies, (3) study selection, (4) charting the data and (5) collating, summarising and reporting the results.[9]

### Stage 1: identifying the research question

The following research questions will be addressed:

#### Primary objectives

1. Which immunotherapies have been, or are currently being, studied in adult patients with sepsis?
2. What are the study characteristics and patient populations of trials on immunotherapy in adult patients with sepsis?
3. How have clinical trials and cohort studies on sepsis used a personalised immunotherapy approach (including clinical phenotypes; immunoprofiling: cell populations, protein biomarkers and/or immune assays), and have these approaches changed over time?

### Stage 2: identifying relevant studies

A comprehensive literature search will be conducted using PubMed, Embase, Cochrane CENTRAL and ClinicalTrials.gov. Inclusion criteria are (1) clinical trials or cohort studies (including case control studies and observational cohorts) (2) investigating immunomodulatory therapies in (3) adult (≥16 years) patients with sepsis. All studies addressing therapies with an established immunomodulatory effect will be included. Studies addressing therapies with a hypothesised immunomodulatory effect (eg, antibiotics, fluids, statins, immuno-nutrition, β-blockers, vasopressors) will be included, but only if the study investigates the immunomodulatory effects of the treatment. After completion of the initial literature search, we will use a snowball approach to find additional studies of interest.[11] All articles must contain full text written in English or Dutch. Exclusion criteria are (1) case studies, (2) animal studies and (3) studies in healthy volunteers. Systematic reviews and meta-analyses will also be excluded, but searched for potentially relevant references.

Search keywords include sepsis, immunotherapy, precision medicine, immunosuppression, anti-inflammatory agents, corticosteroid, anticoagulants, cytokines, granulocyte colony-stimulating factor, immunoglobulins, programmed cell death 1 receptor, Toll-like receptors, polymyxin B, thymosin, protein C, mesenchymal stem cells, antiendotoxin compound and extracorporeal blood removal. Articles retrieved from the database searches will be imported into Rayyan,[12] an open-source programme, to organise and facilitate screening of the search results. The full search strategies can be found in online supplemental methods and online supplemental table 1). Prior to publication the search will be repeated to enable the inclusion of recently published studies.

### Stage 3: study selection

We will approach the study selection systematically using the inclusion and exclusion criteria described above, providing consistency in the decision making regarding article selection. Screening will be performed independently and in duplicate by two reviewers (MAS and NvM). The first step will be to screen all included articles in Rayyan on title and abstract. Disagreements at the trial/abstract stage will be resolved by discussion between the two reviewers. When uncertainty or disagreement remains, a third reviewer (LAvV) will be consulted to make a final decision regarding inclusion of the study. Final inclusion will be determined based on the analysis of the full text of any papers included by either reviewer during title/abstract screening stage. Disagreements at the full-text stage will be resolved in the same manner as described above at the title/abstract stage.

### Stage 4: charting the data

Data to be charted include:
- ► Author(s), year of publication, study location.
- ► Aim(s) of the study.
- ► Study population, inclusion and exclusion criteria, number of patients.
- ► Study design.

- ► The type of immunotherapy and comparator (if any) used.
- ► Main findings and outcome measures, including measured immune profiles.
- ► If a personalised approach was used, and if so, which approach.
- ► Limitations of the study stated by its authors.
- ► Quality of evidence.

The quality of evidence will be critically appraised using the RoB 2 tool (a revised Cochrane risk of bias tool for randomised trials)[13] and the Newcastle Ottawa Quality Assessment Scale for Cohort Studies.[14]

Two reviewers (MAS and NvM) will independently extract data into pre-specified spreadsheets (online supplemental tables 2–5), separating data extracted from randomised controlled trials and observational controlled studies in order to better account for selection bias and confounding factors. At regular intervals, extracted data from the included articles in this framework will be compared in order to ensure interrater reliability. Any disagreements or inconsistencies will be resolved by discussion, or, when uncertainty or disagreement remains, a third reviewer (LAvV) will be consulted.

### Stage 5: collating, summarising and reporting the results

We will present an overview of the reviewed literature in tabular format with descriptive statistics. In addition, when possible, we will analyse treatment groups including (but not restricted to) strategies that modulate excessive inflammation (treatments targeting the innate immune response, complement, immunothrombosis and endothelial dysfunction, pleiotropic drugs with immunomodulatory effects, immunonutrition, supportive treatments with immunomodulatory effects and non-pharmalogical immunomodulatory strategies) and strategies aiming at immune stimulation (immunomodulatory cytokines and growth factors, intravenous immunoglobulins, mesenchymal stem cells and immune checkpoint inhibitors). Furthermore, we will present an overview of the trials currently recruiting or being studied in a supplementary table (online supplemental table 6).

### Patient and public involvement

No patient and/or public involved.

### ETHICS AND DISSEMINATION

As this is scoping review, approval from a medical ethics committee is not required. Once completed, the review will be submitted for publication in a peer-reviewed journal.

### DISCUSSION

This scoping review will provide structured and detailed information on the types of immunotherapies studied in adult patients with sepsis. It will provide insight into how immunomodulatory trials have been conducted, and to what extent a personalised approach to immunotherapy has been used. An overview of potential immunostimulatory and immunosuppressive treatments will be given, highlighting gaps in the field of sepsis treatment, and identifying strategies to study personalised immunotherapies in adult patients with sepsis.

In this review, we will provide background on pathophysiological processes targeted within the investigated therapies. Compared with previous reviews on this topic,[5 7 15–18] which mainly describe current advances in a narrative way, we here use a systematic approach to identify and summarise the evidence for immunomodulatory agents. In addition, the current review will add to the field of sepsis treatment by focusing on the evidence of personalised therapy.

This review will have several methodological strengths. First, as this will be a scoping review, the broad topic of immunotherapy in sepsis in which many different study designs might be applicable can be addressed in a complete and structured manner.[9] Second, we will systematically assess the risk of systematic errors via bias risk assessments, which will provide robustness of our results and conclusions. Third, we will focus on studies that used a personalised approach to immunotherapy, which is proposed to be the future of sepsis research.[19 20]

A potential limitation of this review lies within the heterogeneity between trials. However, this is inherent when studying clinical trials with strict inclusion and exclusion criteria and is probably a good reflection of daily clinical practice in treating sepsis patients. Although inevitable, different criteria for sepsis have been used over time,[2] leading to heterogeneity in the population included. Another limitation lies within the trials using a personalised approach to immunotherapy, since reliable biomarkers selecting the appropriate patient population that may most likely benefit from the treatment investigated, are still being studied.[21 22] However, systematically reviewing the studies investigating personalised immunotherapy will still add knowledge to the future sepsis management.

In conclusion, finding suitable treatment options for sepsis is crucial for improving the patient's outcome. Summarising which immunotherapies have been, and are currently being, studied will highlight the gaps in in the field of immunotherapy in sepsis. This will be essential to improve further knowledge in the treatment of sepsis.

**Author affiliations**
[1]Intensive Care, Amsterdam UMC Location AMC, Amsterdam, The Netherlands
[2]Center for Experimental and Molecular Medicine, Amsterdam UMC Location AMC, Amsterdam, The Netherlands
[3]Medicine, McMaster University, Hamilton, Ontario, Canada
[4]The Guidelines in Intensive Care Development and Evaluation (GUIDE) Group, Research Institute St. Joseph's Healthcare Hamilton, Hamilton, Ontario, Canada
[5]Department of Health Research Methods, Evidence, and Impact, McMaster University, Hamilton, Ontario, Canada
[6]Division of Gastroenterology, McMaster University, Hamilton, Ontario, Canada
[7]Internal Medicine, Radboudumc, Nijmegen, The Netherlands
[8]Radboud Center for Infectious Diseases, Radboudumc, Nijmegen, The Netherlands
[9]Intensive Care, Radboudumc, Nijmegen, The Netherlands

[10] 4th Department of Internal Medicine, National and Kapodistrian University of Athens, Athens, Greece
[11] Internal Medicine, Division of Infectious Diseases, Amsterdam UMC Location AMC, Amsterdam, The Netherlands

**Contributors** MAS, TvdP, WJW, APJV and LAvV formed the study and outlined the proposal. This protocol, including the supplement, was written by MAS, NvM and LAvV. MAS, NvM, JCD, SJWO, MGN, MCAM, APJV and LAvV made a substantive intellectual contribution to the design of the protocol, the study aim and research questions. They jointly developed the search strategy and data extraction framework. MAS, NvM, JCD, SJWO, MGN, PP, EJG-B, MCAM, TvdP, WJW, APJV and LAvV edited the manuscript and approved the final version.

**Funding** MAS, NvM, MGN, PP, EJG-B, TvdP, WJW and APJV are supported by the European Commission (Horizon 2020, ImmunoSep, grant number 847422). WJW is supported by the Dutch Organisation for Scientific Research (Nederlandse Organisatie voor Wetenschappelijk Onderzoek; VIDI grant 91716475). LAvV is supported by the Netherlands Organisation for Scientific Research (VENI grant 09150161910033).

**Disclaimer** The funding sources had no involvement in the study design, in the writing of the protocol and in the decision to submit the article for publication.

**Competing interests** SJWO reports grants and/or contracts from Physician Services Incorporated Ontario and Canadian Institutes of Health Research, and support from the American Thoracic Society and the European Respiratory Society. MGN reports an ERC Advanced Grant, a Spinoza Grant and is on the scientific advisory board of SOBI. PP is on the DSMB of the IL-7 trial 'ILEAD'. EJG-B reports grants and/or contracts from Abbott CH, bioMerieux, MSD, Sobi and ThermoFisher BRAHMS. MCAM is on the DSMB of the PACER trial and the National Dutch Guideline on anticoagulation in COVID-19. TvdP reports project grants from the European Commission and is on the DSMBs of RECAP-CAP, SuDDICU and the CONFIDENT trial. WJW reports a project grant from ZonMW/NWO and receives consulting fees from Merck, Sobi and GlaxoSmithKline. APJV receives consulting fees from InflaRx.

**Patient and public involvement** Patients and/or the public were not involved in the design, or conduct, or reporting, or dissemination plans of this research.

**Patient consent for publication** Not applicable.

**Provenance and peer review** Not commissioned; externally peer reviewed.

**ORCID iDs**
Marleen A Slim http://orcid.org/0000-0002-6281-040X
Joanna C Dionne http://orcid.org/0000-0002-9401-6868
Evangelos J Giamarellos-Bourboulis http://orcid.org/0000-0003-4713-3911

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
