## [Reviewer comments · BMJ Open]

ARTICLE DETAILS

TITLE (PROVISIONAL)	Personalized immunotherapy in sepsis: a scoping review protocol
AUTHORS	Slim, Marleen; van Mourik, Niels; Dionne, Joanna; Oczkowski, Simon; Netea, M; Pickkers, Peter; Giamarellos-Bourboulis, Evangelos; Müller, Marcella; van der Poll, Tom; Wiersinga, Joost; Vlaar, Alexander; van Vught, Lonneke

VERSION 1 – REVIEW

REVIEWER	Rujipat Samransamruajkit Chulalongkorn UNiversity, Pediatric Critical Care
REVIEW RETURNED	01-Feb-2022

GENERAL COMMENTS	This Personalized immunotherapy in sepsis: a scoping review protocol, if it is done carefully review benefit and risk will be benefit to the sepsis management in the future
--

REVIEWER	Andre Kalil University of Nebraska Medical Center
REVIEW RETURNED	30-Mar-2022

GENERAL COMMENTS	The authors describe their plans/protocol for a scoping review on personalized immunotherapy. Introduction is short and objective. However, I ask the authors if before searching for “evidence on immunotherapy in the treatment of sepsis”, shouldn’t they first search for the actual evidence that supports or not the use of immunotherapy? This is relevant because we still have no effective or proven immunotherapy for bacterial sepsis. On page 7, lines 5-9, I suggest to not mix clinical trials with cohort studies, case control, and observational cohorts. The reason for that is because interventional observational studies are plagued by selection bias and confounding. Thus, I recommend two groups of studies to be analyzed separately: 1) randomized controlled trials, and 2) observational controlled studies: case-control and cohort-control. All uncontrolled studies (single-arm, case-series, case reports) should be excluded. The remaining of the methods section is fine. Discussion section is well written, but is missing the description of which will be the limitations of this protocol regarding the immune profiles or signatures. Please add a new sentence on that.
---

VERSION 1 – AUTHOR RESPONSE

Reviewer: 1

Dr. Rujipat Samransamruajkit, Chulalongkorn University

Comments to the Author:

1) This Personalized immunotherapy in sepsis: a scoping review protocol, if it is done carefully review benefit and risk will be benefit to the sepsis management in the future.

We thank the reviewer for his response. We will conduct this review very carefully and hope that our review will contribute to future sepsis care.

Reviewer: 2

Dr. Andre Kalil, University of Nebraska Medical Center

Comments to the Author:

The authors describe their plans/protocol for a scoping review on personalized immunotherapy.

1) Introduction is short and objective. However, I ask the authors if before searching for “evidence on immunotherapy in the treatment of sepsis”, shouldn’t they first search for the actual evidence that supports or not the use of immunotherapy? This is relevant because we still have no effective or proven immunotherapy for bacterial sepsis.

We thank the reviewer for his compliment and agree that it is important to find evidence for the use of immunotherapy in sepsis. This is one of the reasons why we will perform this scoping review. By conducting our review on the evidence available on immunotherapy in sepsis, we will automatically answer the reviewer’s question whether there is any actual evidence available for the use of immunotherapy in sepsis.

2) On page 7, lines 5-9, I suggest to not mix clinical trials with cohort studies, case control, and observational cohorts. The reason for that is because interventional observational studies are plagued by selection bias and confounding. Thus, I recommend two groups of studies to be analyzed separately: 1) randomized controlled trials, and 2) observational controlled studies: case-control and cohort-control. All uncontrolled studies (single-arm, case-series, case reports) should be excluded. The remaining of the methods section is fine.

We agree with the reviewer that randomized controlled trials and observational studies should not be mixed. For this reason, we will separate our results into randomized controlled trials and observational controlled studies. The new sentence concerning the abovementioned change is (page 7 lines 188-189):

Two reviewers (MS, NvM) will independently extract data into pre-specified spreadsheets (Supplementary Table 2 till 5), separating data extracted from randomized controlled trials and observational controlled studies in order to better account for selection bias and confounding factors.

In addition, we also understand that observational studies without controls are prone for selection bias and confounding. However, we would prefer to include them in our study for two reasons:

1. We would like to include phase 1 trials looking into the safety of immunotherapy and those trials are uncontrolled studies.

2. Since our aim is to review all available studies addressing immunotherapy in sepsis, we believe that it would be important to include uncontrolled cohort studies as well, as these studies can lead to additional hypotheses-generating information and may show possible research gaps.

We will pay attention to the possible biases and take this into account in our results.

3) Discussion section is well written, but is missing the description of which will be the limitations of this protocol regarding the immune profiles or signatures. Please add a new sentence on that.

We thank the reviewer for his compliment. Indeed, we did not elaborate on the limitations of looking into trials using a personalized approach to immunotherapy, therefore we added the following sentence (page 9 lines 235-239):

Another limitation lies within the trials using a personalized approach to immunotherapy since reliable biomarkers selecting the appropriate patient population that may most likely benefit from the treatment investigated, are still being studied. However, systematically reviewing the studies investigating personalized immunotherapy will still add knowledge to the future sepsis management.